# Assembly of complex viruses exemplified by a halophilic euryarchaeal virus

Luigi De Colibus[1], Elina Roine [2], Thomas S. Walter [1], Serban L. Ilca [1], Xiangxi Wang[3], Nan Wang[3], Alan M. Roseman[4], Dennis Bamford [2], Juha T. Huiskonen [1,2,5] & David I. Stuart [1,6]

Many of the largest known viruses belong to the PRD1-adeno structural lineage characterised by conserved pseudo-hexameric capsomers composed of three copies of a single major capsid protein (MCP). Here, by high-resolution cryo-EM analysis, we show that a class of archaeal viruses possess hetero-hexameric MCPs which mimic the PRD1-adeno lineage trimer. These hetero-hexamers are built from heterodimers and utilise a jigsaw-puzzle system of pegs and holes, and underlying minor capsid proteins, to assemble the capsid laterally from the 5-fold vertices. At these vertices proteins engage inwards with the internal membrane vesicle whilst 2-fold symmetric horn-like structures protrude outwards. The horns are assembled from repeated globular domains attached to a central spine, presumably facilitating multimeric attachment to the cell receptor. Such viruses may represent precursors of the main PRD1-adeno lineage, similarly engaging cell-receptors via 5-fold spikes and using minor proteins to define particle size.

[1] Division of Structural Biology, University of Oxford, Wellcome Centre for Human Genetics, Oxford OX3 7BN, UK. [2] Research Program in Molecular and Integrative Biosciences, Faculty of Environmental and Biological Sciences, University of Helsinki, 00014 Helsinki, Finland. [3] National Laboratory of Biomacromolecules, CAS Center for Excellence in Biomacromolecules, Institute of Biophysics, Chinese Academy of Sciences, 100101 Beijing, China. [4] Division of Molecular & Cellular Function, School of Biological Sciences, Faculty of Biology, Medicine and Health, Manchester Academic Health Science Centre, University of Manchester, The Michael Smith Building, Oxford Road, Manchester M13 9PL, UK. [5] Helsinki Institute of Life Science HiLIFE, University of Helsinki, 00014 Helsinki, Finland. [6] Diamond Light Source Ltd, Harwell Science & Innovation Campus, Didcot OX11 0DE, UK. Correspondence and requests for materials should be addressed to J.T.H. (email: juha@strubi.ox.ac.uk) or to D.I.S. (email: dave@strubi.ox.ac.uk)

Viruses are pared-down systems, honed by evolution to achieve complexity with genetic economy. Although structurally diverse, there are rather few underlying solutions to the jigsaw-puzzle of assembling a protein container for the genome, reflected in the small number of very different, presumably ancient, structural lineages observed[1]. Capsids of one lineage, which includes adenovirus, bacteriophages such as PRD1 and giant viruses almost as large as a bacterial cell (of which mimivirus was the first identified[2]), are built mainly from many copies of a trimeric major capsid protein (MCP), subunits of which comprise two side-by-side upright β-barrels, with the three subunits arranged to form a pseudo hexamer[3]. Recently, archaeal viruses with related architecture have been identified containing 1620 copies of two smaller (single β-barrel) MCPs[4–6] (VP4 and VP7 for the prototypic $T = 28$ virus, SH1). These MCPs, each resembling half of a double β-barrel, are arranged in hetero-hexameric capsomers with a similar β-barrel configuration to members of the PRD1-adeno lineage[1] surrounding an internal lipid vesicle that encapsulates the dsDNA genome[7]. To understand how such complex systems self-assemble, we have determined the structure of SH1 to 3.8 Å resolution.

SH1 is an icosahedral double-stranded DNA virus with an internal membrane. It infects the halophilic archaeon *Haloarcula hispanica*[8]. At low resolution[5], the ~100 nm diameter $T = 28$ icosahedral virions comprise hexameric capsomers, each decorated with either 2 or 3 turrets (type II and III hexamers respectively), enclosing an underlying lipid vesicle and within that the genome. At each 5-fold symmetric vertex there is a symmetry-mismatch due to attachment of 2-fold symmetric horn-like spikes. Of the 15 structural proteins, 11 (VP1-VP7, VP9, VP10, VP12 and VP13) have been verified by protein chemistry[4,6]. Dissociation studies showed that VP2, 3, 4, 6, 7 and 9 are associated with the protein capsid, and VP1, 5, 10 and 12 with the lipid vesicle[6]. The two MCPs VP4 and VP7 form the crenellated capsomers, some of which (type II) have two and others (type III) three turrets that form the external crenellations[5]. VP2 was proposed to be an elongated fibre-like protein suitable for forming the spikes[4], which also harbour VP3 and VP6[5]. VP12 is the major protein component of the lipid vesicle. SH1 has similarity to *Thermus thermophilus* virus P23-77 for which a low-resolution virion structure and the structure of the two MCPs are known[9,10]. Here, we use the high-resolution structure of SH1 to reveal how this combination of multiple smaller viral jellyroll components, the building blocks of very many viruses, is able, aided by minor capsid proteins, to self-assemble into a PRD1-like capsid, assess the evolutionary links with the PRD1-adeno structural lineage, and describe the architecture of presumed receptor-binding spikes attached at the icosahedral vertices.

## Results

**Structure determination**. SH1 was grown and purified as reported before (Methods, ref.[6]) and the structure determined by cryogenic electron microscopy (cryo-EM). In addition to mature virions, we prepared virions treated with prolonged exposure to sulphate to remove the spikes for early crystallisation experiments (Methods)[5]. For the cryo-EM analysis of the bulk of the structure, data from both sorts of particles were merged and the structure determined using a method accounting for the focus gradient and distortions from icosahedral symmetry in these large particles (Methods, Supplementary Table 1)[11]. This significantly improved the maps (Fig. 1a–c, Supplementary Figs. 1–3), providing a resolution of 3.8 Å[11,12]. The resolution achieved using the merged data was slightly better than when only the larger subset of particles (those with spikes) was used (4.2 Å). Note that in well-ordered regions of the best map bulky side chains are clearly

resolved and the β-strands are well separated (Fig. 1c and Supplementary Fig. 1). As expected, the resolution varies in different parts of the structure and for some less well-ordered regions the connectivity of the protein chain was revealed more clearly in a map where a B-factor (typically 100 Å$^2$) was applied to reduce the high-resolution noise. Multiple copies of the MCPs VP4 and VP7 are present in each icosahedral asymmetric unit and their interpretation was assisted by averaging of the map across these copies (see Methods). Overall, a variety of maps were used in parallel to fully interpret the structure of the capsid excluding the spikes. The structure of the 2-fold symmetric spikes protruding at each of the icosahedral 5-fold axes was analysed using localised reconstruction applied to 3918 spike-containing particles[12]. Firstly, the structure of the whole vertex (without symmetry) was determined at an overall resolution of 7.2 Å (~5 Å in the capsid region but lower than 8 Å within the spike, Supplementary Figs. 2f and 4). To investigate if the poor resolution of the spike was due to the reconstruction locking onto the capsid and blurring the flexible spikes, the reconstruction was repeated for the spike alone (with 2-fold symmetry applied, see Methods for details). This yielded an improved map at 6.2 Å resolution (Fig. 1a and Supplementary Fig. 2e).

**Protein structures**. The quality of the density allowed amino acid sequences to be fitted and structures refined into most of the density (Methods, Supplementary Table 1). The atomic model of the SH1 capsid is shown in Fig. 1d, and the three principle capsid proteins in Fig. 1e–g. The penton protein VP9 (149 residues) surrounds the 5-fold axes and comprises a classic viral β-jellyroll, strikingly similar to the penton protein P31 of PRD1 (75% of the Cα atoms of P31 can be superimposed on VP9 with RMSD 3.6 Å, Supplementary Fig. 5). The major difference between P31 and VP9 is that the first 56 residues of VP9 extend inwards, in a largely helical arrangement to form tight interactions with VP12, which continues inwards to form a bundle of five transmembrane helices around the 5-fold axis (Fig. 1h). The viral membrane itself is not well ordered and our analysis reveals little information beyond that previously reported[5].

The MCPs VP4 (232 residues) and VP7 (185 residues) are arranged as hetero-hexamers (capsomers), underneath which lie, unexpectedly, less well-ordered proteins. The overall structures and arrangement of VP4 and VP7 are broadly as predicted in the light of the structures of the isolated MCPs of the related virus P23-77[10]. VP7 and VP4 comprise upright viral jellyroll structures with VP4 topped by a further small (~75 residue) β-barrel inserted in the DE loop (the loop between the D and E strands of the canonical viral jellyroll), as seen in VP17 of P23-77. The MCPs of SH1 and P23-77 are strikingly similar (80% of residues aligned with 2.8 Å RMSD between VP4 and VP17, and 59% of residues aligned with RMSD 1.8 Å between VP7 and VP16) (Supplementary Fig. 6).

**MCP assembly**. By placing VP4 and VP7 in the context of the mature virion we can probe particle assembly using protein−protein interaction area as an approximation of the strength of the interaction (Supplementary Table 2). It is in principle possible for VP4 and VP7, each with a single cysteine, to form hetero-dimers using disulphide linkages[4]; however, we saw no evidence for such linkages. The smallest building block is a VP7−VP4 heterodimer formed within the hetero-hexamer (Fig. 2a). A six-residue N-terminal extension of VP7 is strapped onto the β-barrel of its clockwise neighbour to stabilise the dimer, if and only if, that subunit is VP4. This VP7−VP4 building block corresponds closely to the double β-barrel structure of a subunit of the P3 MCP in PRD1; indeed, the VP7 C-terminal tail heads towards the

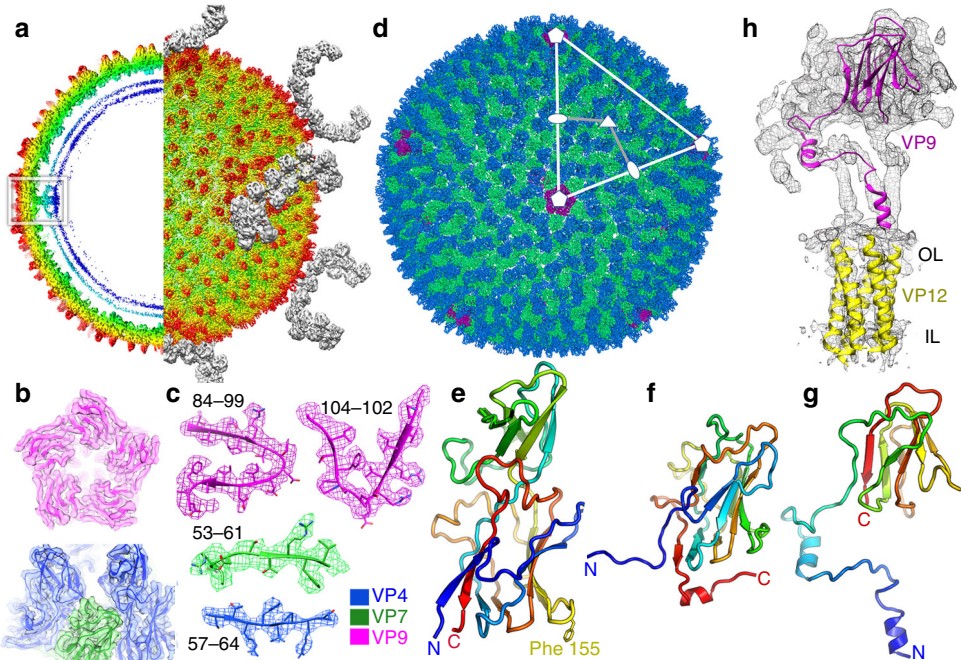

**Fig. 1** SH1 structure overview. **a** Radially coloured SH1 cryo-EM map, a central section is shown to the left and the outer view to the right. Horn-like spikes are shown correctly positioned on the surface in grey. The white inset box marks the 5-fold vertex structure. VP4 turrets are red and the headgroups of the outer and inner leaflets of the membrane are in cyan and blue respectively. **b**, **c** Representative density with fitted structures for VP4 (coloured blue), VP7 (green) and VP9 (magenta), **b** shows an overview and **c** shows close-ups (residue ranges are indicated for each). **d** Overall structure of SH1. Representative 5-fold, 3-fold and 2-fold axes are indicated and the white-grey kite shape delimits an icosahedral asymmetric unit. The outer diameter is ~800 Å. **e** Cartoon representation of VP4, rainbow coloured from the N-terminus (blue) to C-terminus (red). **f** Similar style cartoon representation of VP7. **g** Similar style cartoon of VP9. **h** Cryo-EM map after blurring (with a B-factor of 100 Å$^2$) extracted from the 5-fold vertex region (white box in **a**), showing a subunit of VP9 (magenta) and transmembrane helices of five copies of VP12 fitted to the density (OL outer leaflet, IL inner leaflet)

N-terminus of VP4, which in PRD1 P3 are joined. Furthermore, the P3 β-barrels are asymmetric in a similar fashion to those of the VP7−VP4 pair, and in both PRD1 and SH1 the turret is formed from a small additional barrel. Indeed, it is possible to superimpose the VP7−VP4 pair with P3 and match 122 residues with an RMSD of 4.0 Å (Supplementary Fig. 7). These intracapsomer interactions are likely the lowest level drivers of assembly. The remaining MCP−MCP protein interfaces within a hetero-hexamer are between these VP7−VP4 building blocks and the spare VP7 subunits within some hetero-hexamers (in VP7−VP7 interactions the C-terminal strap is not ordered) (Supplementary Table 2).

The higher-level organisation of the capsid requires the correct selection and orientation of the hexameric capsomers; this comes, at least in part, from lateral interactions between them. Key to this level of assembly are protrusions of VP4 from the perimeter of the hetero-hexamers. These are formed of extended EF loops (that lie between the E and F strands of the canonical viral jellyroll), which bear a hydrophobic tip comprising Phe 155 (Fig. 2a). This hydrophobic peg fits into a corresponding notch formed between the subunits of the adjacent hetero-hexamer. There are three types of such notch, VP4−VP7, VP7−VP4 and VP7−VP7; these are generally well-matching hydrophobic cavities although there may be preferences for particular types of interactions (Fig. 2b). We propose that these specific hydrophobic notches play a central role in dictating correct capsid assembly, as outlined below, and indeed the use of Phe side chain pegs in hydrophobic sockets to control oligomer assembly has been previously observed[13,14]. Additional lateral interactions between hexamers, similar to those predicted for P23-77, also occur (Fig. 2a). Hetero-hexamers neighbouring the 5-fold pentons also utilise the VP4−VP7

notch, in this case engaged by an extended EF loop of VP9, once again ensuring the correct structural register (Fig. 2a). In addition, these hexamers contact an underlying membrane protein, probably, on the basis of prior biochemical evidence, VP12 [4,6]. Elsewhere the membrane lies between ~31 and ~62 Å below the base of the MCPs and does not apparently form direct interactions, and there are no stretches of residues missing from the MCPs substantial enough to form points of membrane attachment. The final set of MCP interactions, with globular proteins nestling under the hexamers, is discussed below.

To understand the higher-level organisation, it is necessary to define the different MCP hexamers that make up the capsid. The extra height of VP4 compared to VP7 means that each VP4 subunit contributes a turret, two in type II hexamers and three in the type III. Type II and type III hexamers have approximate 2-fold and 3-fold symmetry respectively (Fig. 2a). There are two type II hexamers in the icosahedral asymmetric unit, one lying on an icosahedral 2-fold axis, so that each icosahedral asymmetric unit contains half of such a hexamer, we term this the 2-fold hexamer. The second type II hexamer lies adjacent to the 2-fold hexamer (termed 2-fold adjacent). These two hexamers are very similar (RMSD 0.8 Å) (Fig. 2c). There are three type III hexamers in the icosahedral asymmetric unit (Fig. 2c). We term the type III hexamers 5-fold adjacent, 3-fold adjacent and central (Fig. 2c). The type III hexamers are slightly asymmetric, so that if they are rotated by 120° the RMSD between a hexamer and its rotated copy varies between ~1.0 and 2.5 Å. This asymmetry exceeds the variation between the three hexamers (RMSD 0.9 Å (5-fold adjacent on 3-fold adjacent), 1.0 Å (central on 5-fold adjacent) and 1.4 Å (3-fold adjacent on central)).

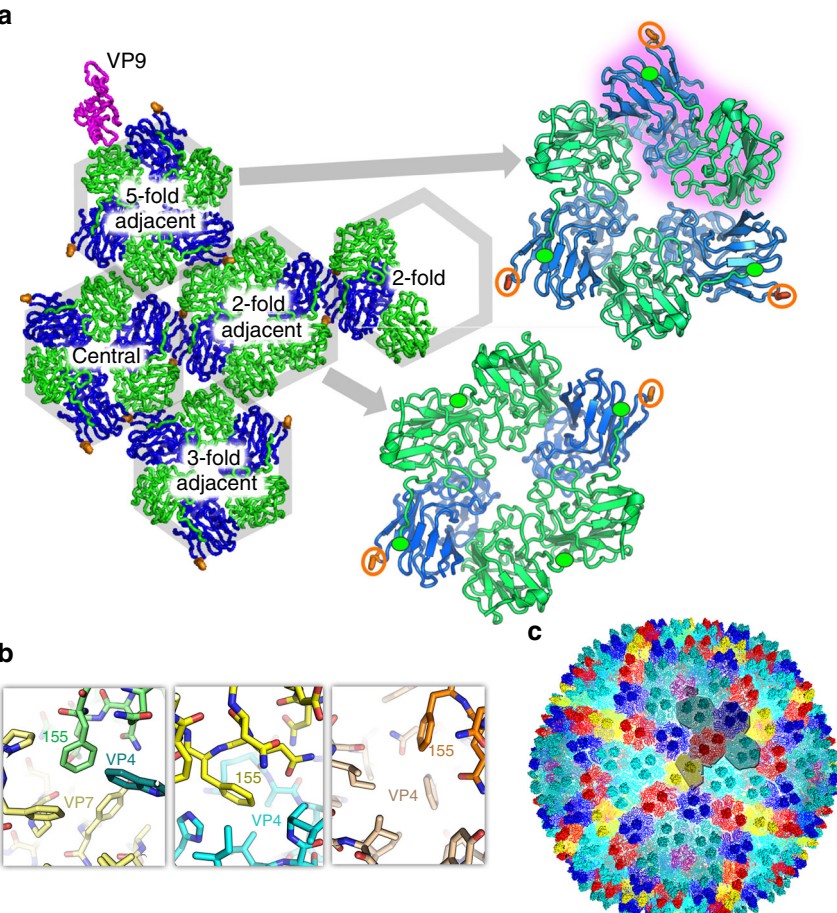

**Fig. 2** Hetero-hexamer building blocks. **a** An icosahedral asymmetric unit, viewed from the inside. VP4 is blue, VP7 green and VP9 magenta. The hexamers are labelled and the arrows show close-ups of a type III turret (upper) with one VP7−VP4 building block highlighted in purple. The N-termini of VP7 are marked by green ellipses and the side chain of Phe 155 of VP4 shown and circled in orange. **b** The three distinct environments of Phe 155 (shown in green, yellow and orange from left to right, with the neighbouring subunits coloured differently). **c** Capsid coloured according to hexamer type: cyan—5-fold adjacent (type III); dark blue—central (type III); pale cyan—3-fold adjacent (type III); red—2-fold adjacent (type II); yellow—2-fold adjacent (type III). VP9 is shown in magenta. One icosahedral asymmetric unit is outlined

**Sub-capsomer organisers**. We believe that the proteins seen unexpectedly attached to the undersides of the capsomers probably play a key role in capsid assembly. The map shows somewhat different structures below each hexamer. We cannot unambiguously identify all of these proteins since the quality of the map does not allow side chain identification (see Fig. 3a). The map is best defined under the 5-fold adjacent type III hexamer and comprises a predominantly helical protein domain of some 80 residues. Secondary structure predictions, knowledge of the minor capsid protein composition[6], their relative mass and the protein size inferred from the density strongly suggest that this protein is VP13 (Methods, Supplementary Fig. 8), and an atomic model has been constructed (Methods, Fig. 3b). Although this protein lies centrally in a depression in the base of a hexamer with quasi 3-fold symmetry it is not 3-fold disordered, instead it makes asymmetric contacts, mostly with two VP7 monomers and one VP4 (Fig. 3a). In contrast, the density under the type II hexamers mirrors the 2-fold symmetry of the hexamers, is less well-defined, and is similar between the two type II hexamers (Fig. 3a). Furthermore, these densities appear similar to a 2-fold averaged version of that seen under the 5-fold adjacent type III hexamers (Fig. 3a and Supplementary Fig. 9). It seems that the deviation from 3-fold symmetry of the 5-fold adjacent hexamer noted

above allows VP13 to attach in a unique orientation within the icosahedral asymmetric unit (although it is not clear how this is achieved), whilst the same protein appears to attach to the 2-fold symmetric type II hexamers in a way that is statistically disordered and so does not follow icosahedral symmetry. The undersides of the remaining two type III hexamers are populated by quite different proteins, which appear to contain both helices and β-sheets (Fig. 3a). To test the similarity of the various structures we cut out the regions of interest and superimposed using a molecular replacement program (program Molrep, which allows one density to be fitted onto another[15]). The type II and 5-fold adjacent structures agreed somewhat; however, the most compelling evidence for their similarity is the visual agreement in the 2-fold averaged density, shown in Supplementary Fig. 9. There is however no significant similarity between the densities below the 3-fold adjacent and central hexamers (cc 0.2); these we consider as different proteins. Similar arguments to those that likely identify VP13 suggest that VP10 might lie underneath the central type III hexamer; however, we are unable to assign the remaining 3-fold adjacent type III associated density to any known protein. These underlying structures likely contribute to the overall organisation of the virus by helping glue the hexamers together and possibly also by bridging hexamers to each other or

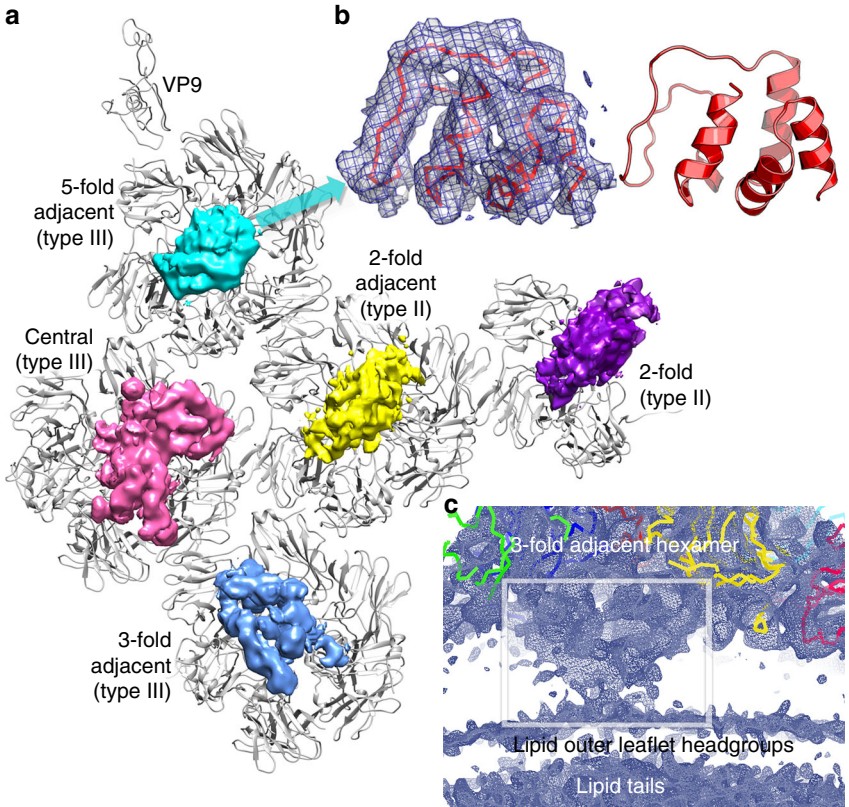

**Fig. 3** Sub-hexamer proteins. **a** View from inside of an icosahedral unit of the capsid. VP9, VP7 and VP4 are shown in grey and the hexamers are labelled. The cryo-EM map shows the density for the sub-hexamer proteins (blurred by the application of a B-factor of 100 Å$^2$). **b** Close-up of density (blurred by the application of a B-factor of 100 Å$^2$) under the 5-fold adjacent hexamer, showing fitted 80 residues of VP13, side view. **c** Lower contour level side view of the density under the 3-fold adjacent hexamer

to the membrane via disordered linkers. Indeed, under most hexamers the membrane structure is somewhat disturbed, especially under the 3-fold adjacent hexamer, consistent with a disordered tether to the membrane (Fig. 3c).

**Capsid growth**. One may speculate how the growth of the virus capsid proceeds, and we propose that it starts with the assembly of robust penton units that associate with the soluble domain of the membrane-associated protein VP12, anchoring them to the membrane. Around these, type III hexamers lock together, kept in register by strong peg-in−hole interactions (two at each hexamer–hexamer interface). We expect that the pool of MCP subunits is largely in the form of free monomers of VP4 and VP7, and VP7−VP4 heterodimers (by analogy with the observations for P23-77 [10]). Good interactions of the Phe 155 peg with adjacent notches would then allow the network of type III hexamers to extend in the correct rotational register, via the icosahedral 3-fold. Type II structures cannot donate as many pegs but are selected around the 2-fold axes where adjacent type II hexamers form pairs of peg-in−notch interactions (Fig. 2). The sub-hexamer proteins likely act as scaffolding proteins, presumably interacting with the membrane and/or each other so ensuring that the virion has exactly the right size. The use of such a scaffold simplifies the assembly problem since there is only one such sub-hexon protein per six MCP subunits and no more than three copies of any sub-hexon protein per icosahedral asymmetric unit.

**Spike structure**. In SH1 and generally in all of the PRD1-adeno lineage viruses[1], it is thought that the structures emerging at the

5-fold vertices bind cellular receptors to initiate infection. In SH1 icosahedral symmetry breaks down in this region, where dimeric horn-like protrusions rise above the surface of the MCP layer at the icosahedral 5-fold axes (Figs. 1a and 4). These structures are less well-resolved than the MCP layer (6.2 Å for a C2 symmetric reconstruction focussed on the horns); however, their architecture is clear. The horns are linked to the capsid via a short (~30 Å) stalk, which is part of the dimeric spike. The arms of the horns initially run parallel to the virus surface before curving up to become perpendicular. The horns span ~211 Å and rise ~182 Å above the surface of the capsid. Each horn comprises ten pearl-like domains arranged in a zig-zag fashion along its rope-like spine (Fig. 4). The pearl-like domains (excluding the terminal two) are extremely similar to each other. Cross-correlation coefficients between individual domains ranged between 0.78 and 0.95, with an average of 0.89 and standard deviation 0.04 (Methods). Adjacent domains are related by a rotation of approximately 180° around the local spine axis and an ~11.5 Å translation along it. The relative orientations may be seen in Fig. 4c–f. Due to the zig-zag arrangement each pearl presents a common face towards the space between the horns. This would fit well with a model for cell attachment whereby the array of common binding faces interacts with a repetitive structure on the cell surface, providing avidity enhanced attachment to the cellular receptor, whose identity remains unknown. The spine to which the pearls are attached is probably made from the elongated VP2 protein (824 residues), a large proportion of which is predicted to be disordered (https://app.strubi.ox.ac.uk/MoreRONN/) and is more than long enough for a single chain to span the complete length of a horn. This is consistent with an extended organiser

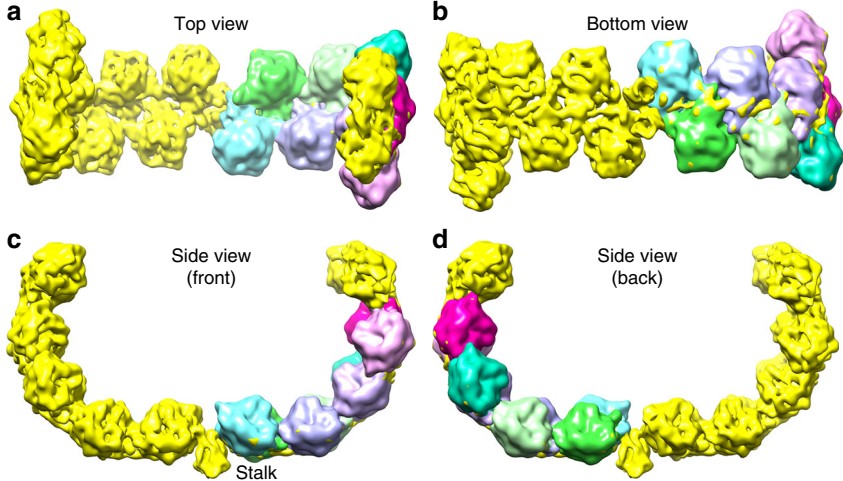

**Fig. 4** The spike. **a–d** show four views. One horn is shown with the raw density (yellow), whilst the other is clothed by eight VP3 pearl-like domains, where the density shown is averaged across all the pearls and the contour level is set to give the correct volume for a protein the size of VP3 (for details see text and Methods). Note that the pearl orientations present a common face to the concave inner surface of the horns, presumed to be the receptor-binding face. The contour level of the map raw density is set at a higher level, to make structural features clearer

molecule (cf. protein P30 of PRD1 [3]). In contrast, the globular pearl domains are almost certainly VP3, since VP3 is the major constituent of the spikes and at 320 residues has an expected volume of ~45,000 Å$^3$, matching that observed (Fig. 4). It is most probable that the domains that cap each horn, and have a different structure, harbour VP6 (230 residues) and contain a substantial contribution from VP2 (possibly including the β-strands predicted to lie towards the C-terminus of VP2).

To understand how the horns attach to the underlying VP9 penton, we inspected an asymmetric localised reconstruction of the vertex (Methods, Supplementary Figs. 4 and 10), which showed contacts with membrane-distal areas in two of the five penton subunits (towards the beginning of strand G, ~residue 113, marked with green stars in Supplementary Fig. 10b). This suggests that the two horns attach to these strands, although the exact interactions were not resolved. In addition, extra density is also associated with the G strand of the adjacent VP9 subunits (marked with yellow stars in Supplementary Fig. 10b) and we suggest that the VP2 tail descends into the penton close to the 5-fold axis and forms a hook to interact with two adjacent subunits, thereby utilising four of the five subunits, presumably by adding a further strand to the exposed strand G of the β-sheet.

## Discussion

Our observation that the first step in SH1 MCP assembly is to form a heterodimer that recapitulates the classic single chain double β-barrel MCP strengthens the evolutionary link of the emerging group of SH1-like viruses with the main double β-barrel virus lineage. SH1 therefore potentially provides a view of a precursor to the hugely extensible double β-barrel family of viruses, which includes the giant viruses. However, unlike some of the double β-barrel viruses (for instance PRD1[3]), where assembly rests on extended minor capsid proteins which cement the hexameric building blocks together and direct the size of the assembly, the single β-barrel viruses use several different minor capsid proteins to attach to a membrane vesicle and then organise two varieties of the hetero-hexameric building blocks, which are locked tightly together via hydrophobic residues in extended loops. The latter is a more complex approach, which may be less extensible to truly giant particles[2]. It is remarkable, however, that the principle of using broadly similar pseudo-hexameric building

blocks to build complexity is preserved across a vast collection of diverse viruses[1], whilst minor proteins determine the exact particle size and varied receptor-binding spikes inserted at the 5-fold axes confer host specificity[1].

## Methods

**Sample preparation**. Haloarchaeal virus SH1 was isolated from a hypersaline water sample originating from Serpentine lake on Rottnest Island, Western Australia[8]. It produces plaques on *Haloarcula hispanica* (ATCC 33960). SH1 particles (1× purified[5]) obtained from infected liquid cultures of *Haloarcula hispanica* were used at the final concentration of 15 mg/ml in modified SH1-buffer (20 mM Tris-HCl pH 7.2, 1 M NaCl and 10 mM MnCl$_2$). Spikeless SH1-VP3VP6 particles were produced by incubation of 1× purified SH1 particles (1 mg/ml) in sulphate buffer (40 mM Tris-HCl pH 7.2, 0.75 M Na$_2$SO$_4$, 40 mM MgSO$_4$) followed by purification by rate zonal centrifugation (80,000 × $g$ for 90 min at 20 °C, Sorvall AH629,) using a linear 5 to 20% sucrose gradient in sulphate buffer. Particles were washed and concentrated by ultrafiltration using modified SH1-buffer.

**Cryo-EM data acquisition and processing**. An aliquot (4 µl) of purified particles (15 mg/ml) was applied to a glow discharged EM grid (C-flat; Protochips) and plunge-frozen in liquid ethane using a vitrification apparatus (Vitrobot mark IV; FEI) operated at 4 °C, 100% relative humidity. Data were acquired using a 300-kV transmission electron microscope (Tecnai F30 Polara; FEI) operated at liquid nitrogen temperature and equipped with an energy filter (GIF Quantum LS; Gatan; zero-loss mode with 20-eV slit width) and a direct electron detector (K2 Summit; Gatan). Movies (22 frames, total electron exposure 22 electrons/Å$^2$) were collected in electron-counting mode using an exposure rate of 8 electrons/pixel/second at ×37,037 magnification, a defocus range of –0.7 to –2.5 µ and a nominal pixel size of 1.35 Å. Data processing was performed within the Scipion software framework[16]. Movie frames were aligned and averaged with dose weighting in MotionCor2 [17] to compensate for specimen drift and mitigate the effects of electron beam-induced specimen damage. Particles were picked from averaged images automatically using ETHAN[18]. Contrast transfer function parameters were estimated locally using CTFIND[19,20]. The initial 3D structure of the full particle was determined in Relion 1.4 [21] using established protocols for image classification and gold-standard structure refinement[22] using a box size of 800 pixels. As a starting model, the previously published structure of SH1 (Electron Microscopy Data Bank (EMDB) accession code EMD-1353) was filtered to low resolution (40 Å) to avoid bias. The resolution of the reconstruction was 4.3 Å, estimated by FSC using a threshold of 0.143. The map was sharpened by applying an inverse B-factor of −160 Å$^2$.

To further improve the resolution, the localised reconstruction method (http://github.com/OPIC-Oxford/localrec)[12] and the related, more recently published block-based technique[11,23] were employed. For the latter, a total of 16,185 particles from 1381 micrographs were selected for two-dimensional alignment and three-dimensional reconstruction using block-based reconstruction. The icosahedral orientation and centre parameters of each particle image determined by Relion[24] were used to guide the extraction of components of three blocks (5-fold-, 5-fold-adjacent- and 2-fold-adjacent-blocks, ~50% bigger than each capsomer) and all three blocks were refined and reconstructed separately using their local defocus

values. In each boxed cryo-EM image, there were 12, 60 and 60 icosahedral-symmetry-related copies for 5-fold-, 5-fold-adjacent- and 2-fold-adjacent-blocks respectively. Knowing the rotation and translational parameters of a virus, the distance $d$ between the centre of one copy in the 3D virus and the centre of the virus along the $Z$ axis (parallel to the incident electron beam) was calculated to solve the gradient in defocus through the capsid. This local defocus of each copy was used to reconstruct the blocks instead of the uniform defocus obtained by fitting. After refinement and reconstruction of the three blocks in Relion[21], the resolution of maps for 5-fold-, 5-fold-adjacent- and 2-fold-adjacent-blocks was improved to 4.0, 4.0 and 4.1 Å respectively; determined by Fourier shell correlation at the 0.143 threshold and testing the effect of the masking used to remove solvent by phase-randomisation (Supplementary Fig. 2a–c). Essentially the same analysis was also carried out using the localised reconstruction method[12]. This analysis focused on the full asymmetric unit (using a total of 235,080 asymmetric units from 3918 particles used to calculate the original 4.3 Å reconstruction). Similar resolution was reached with this approach (4.1 Å; Supplementary Fig. 2d). The block-based maps were further refined in cisTEM[25] leading to an improvement of resolution to 3.8, 3.8, 3.9 Å for 5-fold-, 5-fold-adjacent- and 2-fold-adjacent-adjacent blocks, respectively (Supplementary Fig. 3). Program addup_many_part.py (https://github.com/homurachan/Block-based-recontruction) was used to combine the three blocks into an asymmetric unit. This composite map was used for the final refinement of the capsid structure.

**Localised reconstruction of the spikes.** The localised reconstruction method[12] was used to calculate the spike reconstruction within the Scipion software framework[16]. For analysis of the 2-fold spike alone, partial signal subtraction was used to remove the contribution of the icosahedral capsid from the images. The coordinates of the 12 spike sub-particles were calculated for each particle image and the sub-particles were extracted with a box size of 300 pixels. The extracted sub-particles for each spike were processed separately in Relion 2.1 [21] using standard single-particle, gold-standard protocols[22] unless stated otherwise. A starting model was generated with *relion_reconstruct* without imposing any symmetry and the sub-particles were subjected to 3D classification. Symmetry of the 5-fold vertex was relaxed using a custom version of Relion 2.1 [21] allowing a restricted angular search over only the five equivalent angles in the C5 symmetry group. The resolution was limited to 60 Å in the expectation step, which improved early orientation searches of the spike. A soft edge mask was created for the initial spike density and used for further rounds of 3D classification and refinement of the spike using standard Relion 2.1 [21], local alignments and applying C2 symmetry. The overall resolution of the C2 spike reconstruction was estimated using FSC (0.143 threshold; Supplementary Fig. 2e) within the same soft edge mask and the map was sharpened by applying a B-factor of –255 Å². An asymmetric (C1) reconstruction of the whole vertex was calculated using the sub-particles extracted from the original unsubtracted particles (Supplementary Fig. 2f). This was achieved by assigning the orientation of the spike (determined above by relaxing the local C5 symmetry) to each vertex. To not blur the capsid density, tilting of the spike relative to the capsid was not considered for this map. This resulted in well-resolved contact between the capsid and the spike and less-resolved tips of the spike. The overall resolution of the C1 vertex reconstruction was determined as above and using auto-masking (Supplementary Fig. 2f). Local resolution and a locally filtered map was calculated in Relion (Supplementary Fig. 4). Reconstruction statistics are listed in Supplementary Table 1.

**Spike analysis.** Subunit positions and orientations were identified using Fourier transform-based local-correlation[26] to perform 3D density matching in a complete six-dimensional search (three translation and three orientation parameters). The procedure is similar to previous methods for docking atomic crystal structures into EM maps[27], but uses 3D density objects directly as search motifs, and so does not require an atomic model to fit. An initial density sub-region was selected as a motif to represent a putative subunit, and was correlated with the cryo-EM map using a full six-dimensional local-correlation search, with sampling at 1.35 Å/pixel and 8° rotations, using the ShapEM program (AMR, unpublished). This returned 20 clearly defined peaks, with correlation scores significantly above background or other peaks (cross-correlation coefficients of subsequent peaks were less by more than 20 standard deviations of the cluster of the top 20). Since the map has 2-fold symmetry, ten unique densities were extracted (aligned using the parameters determined in the search) as sub-volumes, examined and averaged. A mask was created to define the molecular boundary of the common region in all the densities, using Chimera[28] for visualisation and Segger[29] to define segments using the watershed method. Segments were manually selected to cover the visually apparent boundary of the subunit. The mask was then smoothed by convoluting with a sphere of constant density with radius corresponding to 10 Å. The ShapEM correlation search was repeated using this more accurate mask and a density model created from the sum of the eight sub-volume densities with the highest correlation coefficients. The threshold for the masking was set so that the volume enclosed was 60,000 Å³, greater than the expected volume of a subunit of 45,000 Å³, thereby extending to include some surrounding solvent region or interfaces. 3D density sub-volumes were extracted into a common orientation using the parameters from the top 20 correlation peaks from the search and were analysed. The inverse transformations were used to place the density model onto the cryo-EM map

corresponding to the peak positions and orientation parameters determined, shown in Fig. 4. For final visual representation in Chimera, the fit-in-map function was used to interpolate the ShapEM fits to a local optimal position. Movements were less than 6° and 1 pixel. To calculate the cross-correlation coefficients between the raw densities representing each subunit, a local search of every individual volume vs. every other was performed using the extracted aligned sub-volumes, with 4° steps, also in ShapEM. Adjustments to match their orientations optimally were less than or equal to 8°.

**Model building, refinement and analysis.** Atomic models of VP4 and VP7 MCPs generated by I-Tasser[30,31] were rebuilt into the cryo-EM density using Chimera[28] and Coot[32]. The portion of the viral asymmetric unit (AU) composed by the MCPs was assembled by fitting 12 copies of VP4 (residues 7−232), 15 copies of VP7 (residues 2−176). Subsequently, the density corresponding to the volume of these molecules within the AU was extracted with Phenix suite phenix.map_box[33] (using a soft mask and atom mask radius around the atoms of 4 Å). To facilitate the sequence assignment, the map features were improved by B-factor sharpening and blurring in both REFMAC[34,35] and PHENIX[33,36]. In addition, two sets of non-icosahedral symmetry (NIS) operators relating the 12 copies of VP4 and 15 copies of VP7 respectively were calculated with phenix.simple_ncs_from_pdb from the coordinates of the fitted monomers[33,37] and these operators were used to perform NIS density averaging with phenix.ncs_average (mask radius = 4 Å)[33,37]. In many regions the averaged density was easier to interpret (Supplementary Fig. 11). The atomic models of single monomers of VP4 and VP7 were built using Coot[32] and subsequently rebuilt and refined in Rosetta release version 2018.09.60072 using protocols optimised for cryo-EM maps[38] using -denswt = 40. The best-scoring model as estimated by density fit and geometry was selected and used in Coot[32] to guide further model building and optimisation before replicating onto the other NIS copies to reconstitute the AU[37]. A similar strategy was applied to build the atomic model of the minor protein VP9 located at the 5-fold icosahedral axis. Regions of the single protomer where the density was difficult to interpret were removed from the model and rebuilt with Rosetta-ES[39] using -denswt = 40. The residues (14–28) at the N-terminus of VP9 have been modelled as polyalanine. The cryo-EM density corresponding to VP13 was highly compact, showing clearly the presence of tubes of density characteristic of helices, which became better defined after blurring in REFMAC[34,35] with a B-factor of 100 Å². This map allowed the placement of approximately 80 residues. The polyalanine atomic model was built with Coot[32] and relaxed into the density with Rosetta[40]. Secondary structure predictions were performed with the PSIPRED server (http://bioinf.cs.ucl.ac.uk/psipred/) on the minor capsid proteins to search for the sequence with a number of secondary structure elements similar to those identified in the map. The only sequence with mainly helical secondary structure and a suitable number of residues was VP13 (Supplementary Fig. 8). The transmembrane region (27 residues long) of VP12 was built as polyalanine model and rigid body fitted using Coot[32] into the cryo-EM density blurred in REFMAC[34,35] with a B-factor of 100 Å². The 12 copies of VP4, the 15 copies of VP7 and one copy of VP9 comprising the AU were refined together with the neighbouring chains in phenix.real_space_refine[33,41] using non-icosahedral constraints, secondary structure restraints and reciprocal space B-factor refinement. This strategy was applied to prevent clashes at the interface of the viral AUs. The complete AU was then assembled by combining the refined 12 copies of VP4 (residues 7−232), 15 copies of VP7 (residues 2−176), one copy of the minor capsid protein VP9 (residues 14−148), the transmembrane portion of VP12 and VP13 protein. Subsequently, the icosahedral symmetry operators were found in the full particle cryo-EM map with phenix.map_symmetry[33]. The complete AU was placed in full particle cryo-EM map with Coot[32] and the full particle atomic model was generated by applying the icosahedral symmetry to it with phenix.apply_ncs[33], followed by rigid body refinement of the 60 copies of the AU in phenix.real_space_refine[33,41]. All refinement steps were performed in the presence of hydrogen atoms. This model was validated by Molprobity[42], EMringer[43] and CaBLAM[44]. To analyse the apparent 2-fold disorder of VP13 under certain hetero-hexamers, density corresponding to the VP13 protein beneath the 5-fold adjacent, 2-fold adjacent and 2-fold symmetric hexamers were extracted by fitting the VP13 atomic model to the map (in the latter two cases after fitting a single copy of VP13 a 2-fold related copy was generated to complete the fit to the density around the local 2-fold of the hexamer), using Coot[32] and phenix.map_box[33] (using a soft mask and atom mask radius around the atoms of 4 Å). Extracted densities beneath the 5-fold adjacent and 2-fold adjacent hexamers were then averaged using the 2-fold NCS in the atomic model in Coot[32] (Supplementary Fig. 9). The densities beneath the remaining two type III hexamers were extracted with phenix.map_box[33] (using a soft mask and atom mask radius around the atoms of 6 Å) and the comparison between them was performed with MOLREP[15] employing the protocol to fit a model into a density but instead of using an atomic model of the protein of interested we used its cryo-EM density.

**Reporting summary.** Further information on experimental design is available in the Nature Research Reporting Summary linked to this article.

## Data availability

The atomic coordinates of SH1 have been submitted to the Protein Data Bank under accession code 6QT9. The cryo-EM maps have been deposited at the Electron Microscopy Data Bank under codes EMD-4633, EMD-4634 and EMD-4656. The data that support the findings of this study are available from the corresponding authors upon request.

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

## Acknowledgements

The work was supported by the UK Medical Research Council (grant MR/N00065X/1 to D.I.S.) and the European Research Council under the European Union's Horizon 2020 research and innovation programme (649053 to J.T.H.), S.L.I. by a Wellcome Trust Studentship (109135/Z/15/A) and L.D.C. by a WHO/Gates foundation award (RG. IMCB.I8-TSA-083). The work of the Wellcome Trust Centre in Oxford is supported by a Wellcome Trust core award 090532/Z/09/Z. The OPIC electron microscopy facility was founded by a Wellcome Trust JIF award (060208/Z/00/Z) and is supported by a Wellcome Trust equipment grant (093305/Z/10/Z). The Wellcome Trust, MRC and BBSRC also support the UK national cryo-EM facility (eBIC), which enabled provision of the K2 detector. D.I.S. is a Jenner Investigator. The project originated as an Instruct Collaboration (now Instruct-ERIC). We thank University of Helsinki for the support to EU ESFRI Instruct Centre for Virus Production and Purification used in this study. This work was supported by Chinese Academy of Medical Sciences Oxford Institute and by Academy Professor (Academy of Finland) funding grants 255342 and 256518 to D.B. The authors thank Sari Korhonen for excellent technical assistance, Jun Dong, Robert Esnouf and Jonathan Diprose for IT support, Dr. Tom Terwilliger, Dr. Pietro Roversi, Dr. Pavel Afonine, Oleg Sobolev and Brandon Frenz for helpful discussion on density averaging, model building and refinement with cryo-EM map, and the OPIC electron microscopy facility for technical support.

## Author contributions

L.D.C. contributed to experimental design, performed data collection and analysis, E.R. made the sample, T.S.W. performed data collection, S.L.I., X.W., N.W. and A.M.R. performed data analysis, D.B. contributed to project design, J.T.H. and D.I.S. contributed to experiment design and analysis. All authors contributed to writing the paper.

## Additional information

**Competing interests:** The authors declare no competing interests.

