## [Peer Review File · Nature Communications]

Reviewers' Comments:

Reviewer #1:

Remarks to the Author:

This manuscript presents cryo-EM studies of a PRD1-like virus, SH1. The low resolution cryo-EM reconstruction of SH1 was reported by Butcher and colleagues in 2008. Taking advantage of direct electron detectors in the cryo-EM field, now Stuart and colleagues were able to reach near-atomic resolution for the capsid region, and then build atomic models for the MCPs and pentons. In addition to that, they observed some densities for proteins beneath the type-II and type-III hexamers at low resolution. Unfortunately, they couldn't build atomic models for most of those regions. They also tried to reconstruct the vertex complex (spike structure) by applying a C2 symmetry and NCS averaging. Altogether, this paper provides some new insights to the community that is interested in virus assembly. The structural studies appear to be well designed and executed, but there are some major issues that must be addressed before being published in Nature Communications.

Major points:

(1) The authors claimed that they have determined the structure of SH1 to 3.8 Å resolution, however:

A) From the maps the authors showed (fig-1c, supfig-1) it does not look like a typical 3.8 Å map. In a typical 3.8 Å map, one will see clear separation of beta-strands and many bulky side chains, which I did not see here.

B) Most of the half-map FSCs (reflecting different small units of the virus) provided by the authors didn't reach 3.8 Å, except for the FSCs for two small regions, -pen and -unit1 as shown in supfig-3. But those 2 FSC plots do not go to zero at high frequency and negative values are not shown, suggesting there are some masking artifacts during this calculation (such that the masks have correlations). The authors even estimated the resolution to a hundredth of an Angstrom. This level of precision makes no sense in cryo-EM.

C) It is known in the field that half-maps FSC can be a poor indicator for accurate estimation of cryo-EM resolution. So for the regions where atomic models were built, the authors need to provide model:map FSC, real-space CC and d99 (Afonine et al 2018, Acta Crystallographica Section D).

(2) I am very confused about the vertex complex. The authors claimed that the half-map FSC for the vertex complexes is 7.2 Å with C1 symmetry and 6.2 Å with C2 symmetry. How was the FSC calculated? Does this estimation only include the spike structure part, or does it include the capsid as well? If this only includes the spike part, why in supfig-4 is the resolution for the spike part only 9 Å? If this half-map also includes the capsid part, it makes no sense because the capsid region will dominate the FSC calculation. After all, from fig-4 and supfig-4, there is no way that the vertex region is a 6.2 Å map. It appears that the consistency of two maps is being used to overestimate the resolution.

(3) In fig3:

A) The authors state that the densities beneath type-II hexamers look like a 2-fold averaged version of putative VP13. I do not understand this. Why would this be 2-fold averaged? This part should not be affected by applying icosahedral symmetry, because it is completely within the asymmetric unit.

B) The authors state that the densities beneath type-II are also likely VP3 because those densities somewhat agreed with a correlation coefficient of 0.3. Two random maps could have a similar CC. The authors need to analyze those maps more carefully and rewrite this part.

Minor points:

(4) They need to list the refinement statistics of the atomic models somewhere in the extended data, including clash score, Ramachandran, rotamer, RMS deviations etc.

Reviewer #2:

Remarks to the Author:

Colibus et al. report the cryo-EM structure of an archaeal virus SH1 capsid at about 3.8-7 angstrom resolution (major capsid proteins (MCPs) at about 3.8 angstrom and a distinctive twofold spike at about 6 angstrom). While the minor proteins under the MCP are not well resolved. Based on the cryo-EM density map, the authors are able to build atomic models of the two MCPs and trace the backbone of the capsid protein VP13 under the MCP hexamers. It is interesting to see that the atomic model of the SH1 MCP VP7-VP4 heterodimer topologically resembles the counterpart MCP in the bacteriophage PRD1, and they both have double beta-barrels. Based on the observations, the authors propose a SH1 capsid assembly process and suggest that the SH1 is a precursor to the double-barrel family of viruses.

The manuscript will be of specific interest to readers who study viruses in the PRD1-adenoviral lineage as well as other structural virologists. I hope my suggestions below could enhance the paper.

Major points:

P4, L68: Not only the low resolution structure of virus P23-77, but also the atomic structures of the virus P23-77 MCPs are known. See Ref.14 in the main text.

P4, L80-83: The authors should be cautious with merging image data from two sorts of particles for a reconstruction. Is there any region of the merged map that becomes blurred compared with those of the two maps reconstructed independently?

L84: The authors could try to average the 12 copies of VP4 and the 15 copies of VP7 within the asymmetric unit to improve their respective resolutions.

P4, L86: It is unclear what this sentence means.

P6, L135: The term "EF loop extension" should be explained.

P7, L147: A reference is needed for the biochemical evidence.

P7, L148: No detectable interactions between the MCPs and the membrane may not exclude the possibility there are asymmetric interactions, because 1) icosahedral average was applied, and 2) the structural resolution is limited.

P7, L153: The section "overall capsid organization" might be moved up or alternatively incorporated into the section "MCP Assembly".

P8, L176-179: I'm not convinced. It would be helpful if the authors could describe how the secondary structure predictions suggest that the density is VP13, maybe this could be included in supplementary materials.

L180: Any explanation why this is not threefold disordered?

L181-183: Do the authors mean that the underlying minor proteins are less well-defined because

these proteins could adapt the two twofold symmetrical orientations randomly under the twofold type-II hexamers?

L183-184: Could the authors apply twofold average to the density under the 5-fold adjacent type-III hexamer to simulate the less well-defined densities under the type-II hexamers?

L188: It is necessary to add a reference and a brief description for program Molrep.

P11, L154: I expected the authors to discuss the structural differences or biological story of the two sorts of particles. For example, does the "stalk" belong to the main capsid or the spike? What's the purpose of taking time to prepare the sulphate-treated particles?

Minor points:

P3, L63: redundant definition of abbreviation "MCPs";

L63-65: A citation of the Ref. 7, in which the two distinct hexamer types were defined, is needed here.

P4, L83: What are "standard methods"? The authors referenced the extended data Fig.1, while only the focus gradient correction was indicated in the figure legend.

P5, I92: I do not see the various resolutions from the extended data Fig. 3. Should it be extended data Fig. 4?

P5, L96: Extended Data Table 1 is not an atomic model refinement statistics table.

Reviewer #3:

Remarks to the Author:

The viruses in PRD1-adenoviral lineage use trimers of major capsid protein with double jelly-roll fold to form hexamer and use five minor capsid proteins with single jelly-roll structure to form pentamer. The icosahedral capsid of these viruses is formed by 12 pentamers and hundreds or thousands hexamers which depending on the T numbers. The double jelly-roll major capsid protein was widely found in plant RNA virus, such as cowpea mosaic virus and PBCV-1, Bacterial phage, such as PRD1 and PM2, archaeal virus STIV, vaccinia virus and even "virophage" Sputnik. Here, Colibus et al., solved a membrane containing archaeal virus at resolutions of 3.8 Å. The major capsid protein of this virus is single jelly-roll structure. The pentamer is formed by five vp9 proteins with single jelly-roll structure. 12 pentamers and 270 hexamers together form this T = 28 icosahedral virus. The unknown minor proteins attached to the undersides of the capsomers probably assist capsid assembly. This is an interesting story about how single jelly roll protein can be organized as major capsid protein to form an icosahedral virus capsid. The map quality is good. The result and discussion is reasonable.

I only have one minor comment

As far as I know, almost all other double jelly roll viruses have minor capsid proteins that connects the neighboring hexamers. The simplest one, such as sputnik has a "helix" from a unknown minor protein that connects the neighboring hexamers. As authors already mentioned that the minor capsid proteins for double jelly roll virus may help to maintain the T number. However, sometimes minor proteins also

has the function such as enhancing the interaction between capsomers. Is it possible that the single jelly-roll capsomer-capsomer interaction can be stronger than that of double jelly-roll capsomers?

Reviewer #1 (Remarks to the Author):

1) The authors claimed that they have determined the structure of SH1 to 3.8 Å resolution, however:

A) From the maps the authors showed (fig-1c, supfig-1) it does not look like a typical 3.8 Å map. In a typical 3.8 Å map, one will see clear separation of beta-strands and many bulky side chains, which I did not see here.

We thank the reviewer for pointing this out, which prompted us to re-examine the images. We realized that the density was displayed at a rather low contour level, which resulted in the map appearing less well-defined. We have adjusted the contour level and orientation, and now most parts of the 3.8 Å maps (corresponding to the core domains of VP4, VP7 and VP9) show clear separation of beta-strands and many bulky side chains, while the densities for the N-terminus of VP9 and the most exterior loops of VP4 are less ordered. Cryo-EM maps of blocked units and local electron scattering potential maps which illustrate sidechain features are provided in Fig.1c.

B) Most of the half-map FSCs (reflecting different small units of the virus) provided by the authors didn't reach 3.8 Å, except for the FSCs for two small regions, -pen and -unit1 as shown in supfig-3. But those 2 FSC plots do not go to zero at high frequency and negative values are not shown, suggesting there are some masking artefacts during this calculation (such that the masks have correlations). The authors even estimated the resolution to a hundredth of an Angstrom. This level of precision makes no sense in cryo-EM.

Thank you for pointing this out. Indeed, normal reconstruction strategies yield a reconstruction at ~4.3 Å resolution for such a large particle by imposing an icosahedral symmetry. There are two aspects of the analysis that introduce that resolution limit. One is the fact that the complex architecture, and strain imposed by freezing mean that the particles do not strictly conform to icosahedral symmetry; the other is the gradient in defocus through the capsid. To address both of these, a block-based reconstruction method was applied, in which the structure is divided into blocks and a local search for better rotation and translation parameters of each block is performed followed by a further round of 3D classification to allow some plasticity in the structure. At this point the resolution as determined by Fourier shell correlation was 4.0 Å in Relion and 3.8 Å in cisTEM at the 0.143 FSC threshold. Moreover, we now provide the full range of values of the block-based cisTEM FSC plots (supplementary fig-3). The updated figure shows that these 3 FSC plots from cisTEM go to zero, even negative values at high frequency. Note that the black line for the combined particle map does indeed not reach zero at high frequency. We speculate that this might arise from this map being artificially combined from individual blocks.

(2) I am very confused about the vertex complex. The authors claimed that the half-map FSC for the vertex complexes is 7.2 Å with C1 symmetry and 6.2 Å with C2 symmetry. How was the FSC calculated? Does this estimation only include the spike structure part, or does it include the capsid as well? If this only includes the spike part, why in supfig-4 is the resolution for the spike part only 9 Å? If this half-map also includes the capsid part, it makes no sense because the capsid region will dominate the FSC calculation. After all, from fig-4 and supfig-4, there is no way that the vertex region is a 6.2 Å map. It appears that the consistency of two maps is being used to overestimate the resolution.

We recognise that our description was poor and are glad for the opportunity to clarify. The C1 map of the vertex includes both the spike structure (or 'horn') as well as some of the capsid as shown in Extended Figure 4. We agree that the capsid region dominates this calculation and hence we have calculated the local resolution of this map, shown in Extended Figure 4 (calculated in Relion using local FSC), which varies from ~5.5 Å (for the best parts of the capsid) to 9 Å and lower (for the most flexible parts of the spike). This resolution range is consistent with the overall resolution figure of 7.2 Å.

The resolution of the spike is especially low in the C1 vertex map because it is flexible relative to the capsid. To deal with this flexibility, we have also reconstructed a map of the spike alone, imposing C2 symmetry. The overall resolution of this C2 map of the spike ('horn') is 6.2 Å as shown in Extended Figure 2e (FSC=0.143 calculated from two independent half-maps from 'gold-standard' refinement). To get this higher resolution structure of the spike, we used images where capsid contribution had been subtracted and allowed spike particles to rotate relative to the capsid to account for their flexibility as explained in the methods.

We have expanded the text to clarify these points.

3) In fig3:

A) The authors state that the densities beneath type-II hexamers look like a 2-fold averaged version of putative VP13. I do not understand this. Why would this be 2-fold averaged? This part should not be affected by applying icosahedral symmetry, because it is completely within the asymmetric unit.

Thanks for pointing this out. The local 2-fold symmetry arises from their local environment. They sit under type II hexamers which possess 2-fold symmetry and so it appears that the underlying protein can attach either way round, so that we see an average. We have clarified this point in the text and provide figures to back up the assertion (see below).

B) The authors state that the densities beneath type-II are also likely VP3 because those densities somewhat agreed with a correlation coefficient of 0.3. Two random maps could have a similar CC. The authors need to analyze those maps more carefully and rewrite this part.

We have done this, we show 2-fold averaged data in Extended Data Fig.9, to let the reader assess the strength of our claims.

Minor points:

(4) They need to list the refinement statistics of the atomic models somewhere in the extended data, including clash score, Ramachandran, rotamer, RMS deviations etc.

We thank the reviewer for pointing this out. A complete refinement table is now provided as Extended Data Table 1b.

Reviewer #2 (Remarks to the Author):

Colibus et al. report the cryo-EM structure of an archaeal virus SH1 capsid at about 3.8-7 angstrom resolution (major capsid proteins (MCPs) at about 3.8 angstrom and a distinctive twofold spike at about 6 angstrom). While the minor proteins under the MCP are not well resolved. Based on the cryo-EM density map, the authors are able to build atomic models of the two MCPs and trace the backbone of the capsid protein VP13 under the MCP hexamers. It is interesting to see that the atomic model of the SH1 MCP VP7-VP4 heterodimer topologically resembles the counterpart MCP in the bacteriophage PRD1, and they both have double beta-barrels. Based on the observations, the authors propose a SH1 capsid assembly process and suggest that the SH1 is a precursor to the double-barrel family of viruses.

The manuscript will be of specific interest to readers who study viruses in the PRD1-adenoviral lineage as well as other structural virologists. I hope my suggestions below could enhance the paper.

Major points:

P4, L68: Not only the low resolution structure of virus P23-77, but also the atomic structures of the virus P23-77 MCPs are known. See Ref.14 in the main text.

We have now added this to the text.

P4, L80-83: The authors should be cautious with merging image data from two sorts of particles for a reconstruction. Is there any region of the merged map that becomes blurred compared with those of the two maps reconstructed independently?

Agreed, additional explanation is provided in the text, with a justification of the use of merged data. Overall a series of maps were available and were consulted in parallel during the model building.

L84: The authors could try to average the 12 copies of VP4 and the 15 copies of VP7 within the asymmetric unit to improve their respective resolutions.

Agreed. This was performed, as was hinted at in the methods section, we now provide additional images showing the difference in the electron scattering potential maps before and after averaging (Extended Data Fig. 10) and clarify the main text of the paper and the Methods section.

P4, L86: It is unclear what this sentence means.

We have modified the sentence to be clearer.

P6, L135: The term "EF loop extension" should be explained.

A definition is now provided in the text the first time jellyroll strand IDs are used to identify loops.

P7, L147: A reference is needed for the biochemical evidence.

The reference has been added to the text.

P7, L148: No detectable interactions between the MCPs and the membrane may not exclude the possibility there are asymmetric interactions, because 1) icosahedral average was applied, and 2) the structural resolution is limited.

This is true, we slightly modified the text

P7, L153: The section "overall capsid organization" might be moved up or alternatively incorporated into the section "MCP Assembly".

We have merged this into the MCP Assembly section as suggested

P8, L176-179: I'm not convinced. It would be helpful if the authors could describe how the secondary structure predictions suggest that the density is VP13, maybe this could be included in supplementary materials.

As suggested additional info has been provided in the supplementary materials and in Methods.

L180: Any explanation why this is not threefold disordered?

No, I am afraid we don't know. We have indicated this in the text.

L181-183: Do the authors mean that the underlying minor proteins are less well-defined because these proteins could adapt the two twofold symmetrical orientations randomly under the twofold type-II hexamers?

Yes, we have clarified this.

L183-184: Could the authors apply twofold average to the density under the 5-fold adjacent type-III hexamer to simulate the less well-defined densities under the type-II hexamers?

Indeed this was done and is now presented.

L188: It is necessary to add a reference and a brief description for program Molrep.

Reference has been added. Molrep is a standard molecular replacement program, and we feel that details are not necessary.

P11, L154: I expected the authors to discuss the structural differences or biological story of the two sorts of particles. For example, does the "stalk" belong to the main capsid or the

spike? What's the purpose of taking time to prepare the sulphate-treated particles?

The reasons were historical (the project originally used crystallography, for which flexible spikes were fatal), this is irrelevant to the present paper. The stalk belongs to the spike and we hope this is now clear in the modified text.

Minor points:

P3, L63: redundant definition of abbreviation "MCPs";

Corrected.

L63-65: A citation of the Ref. 7, in which the two distinct hexamer types were defined, is needed here.

Citation added.

P4, L83: What are "standard methods"? The authors referenced the extended data Fig.1, while only the focus gradient correction was indicated in the figure legend.

Thank you for pointing this out, we hope we now provide sufficient explanation.

P5, I92: I do not see the various resolutions from the extended data Fig. 3. Should it be extended data Fig. 4 ?

Corrected.

P5, L96: Extended Data Table 1 is not an atomic model refinement statistics table.

Corrected, complete refinement table is now provided as ED Table 1b.

Reviewer #3 (Remarks to the Author):

The viruses in PRD1-adenoviral lineage use trimers of major capsid protein with double jelly-roll fold to form hexamer and use five minor capsid proteins with single jelly-roll structure to form pentamer. The icosahedral capsid of these viruses is formed by 12 pentamers and hundreds or thousands hexamers which depending on the T numbers. The double jelly-roll major capsid protein was widely found in plant RNA virus, such as cowpea mosaic virus and PBCV-1, Bacterial phage, such as PRD1 and PM2, archaeal virus STIV, vaccinia virus and even "virophage" Sputnik. Here, Colibus et al., solved a membrane containing archaeal virus at resolutions of 3.8 Å. The major capsid protein of this virus is single jelly-roll structure. The pentamer is formed by five vp9 proteins with single jelly-roll structure. 12 pentamers and 270 hexamers together form this T = 28 icosahedral virus. The unknown minor proteins attached to the undersides of the capsomers probably assist

capsid assembly. This is an interesting story about how single jelly roll protein can be organized as major capsid protein to form an icosahedral virus capsid. The map quality is good. The result and discussion is reasonable.

I only have one minor comment

As far as I know, almost all other double jelly roll viruses have minor capsid proteins that connects the neighboring hexamers. The simplest one, such as sputnik has a "helix" from a unknown minor protein that connects the neighboring hexamers. As authors already mentioned that the minor capsid proteins for double jelly roll virus may help to maintain the T number. However, sometimes minor proteins also has the function such as enhancing the interaction between capsomers. Is it possible that the single jelly-roll capsomer-capsomer interaction can be stronger than that of double jelly-roll capsomers?

We thank the reviewer for this – indeed that was the implication of the P23-77 MCP structures we published earlier, we now more explicit about the strength of the interactions in the final discussion.

Reviewers' Comments:

Reviewer #1:

Remarks to the Author:

The authors have, in general, addressed all of the concerns that I raised. As a result, the manuscript is significantly improved. There are two minor points (only one of which requires a change in the paper).

- 1) They have still included a correlation coefficient of 0.3 as indicating significant similarity. Apparently, they must believe that only a cc of 0.0 would be indicative of no similarity. But in practice one can easily obtain this type of correlation between two unrelated structures at this resolution. The authors should convince themselves of this by actually trying such comparisons. I therefore strongly suggest that they drop this from the paper.
- 2) In the response to the reviewers (but not the paper) they appear to suggest that an FSC going below 0.0 beyond the resolution limit is actually better than one at 0.0. In fact, such negative correlations typically arise from mask artifacts.

Reviewer #2:

Remarks to the Author:

The authors have adequately addressed my concerns.

REVIEWERS' COMMENTS:

Reviewer #1 (Remarks to the Author):

The authors have, in general, addressed all of the concerns that I raised. As a result, the manuscript is significantly improved. There are two minor points (only one of which requires a change in the paper).

1) They have still included a correlation coefficient of 0.3 as indicating significant similarity.

Apparently, they must believe that only a cc of 0.0 would be indicative of no similarity. But in practice one can easily obtain this type of correlation between two unrelated structures at this resolution. The authors should convince themselves of this by actually trying such comparisons. I therefore strongly suggest that they drop this from the paper.

Done

2) In the response to the reviewers (but not the paper) they appear to suggest that an FSC going below 0.0 beyond the resolution limit is actually better than one at 0.0. In fact, such negative correlations typically arise from mask artifacts.

This does not affect the paper

Reviewer #2 (Remarks to the Author):

The authors have adequately addressed my concerns.